# Omicron variant neutralizing antibodies following BNT162b2 BA.4/5 versus mRNA-1273 BA.1 bivalent vaccination in patients with end-stage kidney disease

Kevin Yau [1,2], Alexandra Kurtesi[3], Freda Qi[3], Melanie Delgado-Brand[3], Tulunay R. Tursun[3], Queenie Hu[3], Miten Dhruve[4], Christopher Kandel[5], Omosomi Enilama [6], Adeera Levin[7], Yidi Jiang[8], W. Rod Hardy[3], Darren A. Yuen[9], Jeffrey Perl[9], Christopher T. Chan[2], Jerome A. Leis[10], Matthew J. Oliver[1,11], Karen Colwill [3], Anne-Claude Gingras [3,12] & Michelle A. Hladunewich[1,11] ✉

Neutralization of Omicron subvariants by different bivalent vaccines has not been well evaluated. This study characterizes neutralization against Omicron subvariants in 98 individuals on dialysis or with a kidney transplant receiving the BNT162b2 (BA.4/BA.5) or mRNA-1273 (BA.1) bivalent COVID-19 vaccine. Neutralization against Omicron BA.1, BA.5, BQ.1.1, and XBB.1.5 increased by 8-fold one month following bivalent vaccination. In comparison to wild-type (D614G), neutralizing antibodies against Omicron-specific variants were 7.3-fold lower against BA.1, 8.3-fold lower against BA.5, 45.8-fold lower against BQ.1.1, and 48.2-fold lower against XBB.1.5. Viral neutralization was not significantly different by bivalent vaccine type for wild-type (D614G) ($P = 0.48$), BA.1 ($P = 0.21$), BA.5 ($P = 0.07$), BQ.1.1 ($P = 0.10$), nor XBB.1.5 ($P = 0.10$). Hybrid immunity conferred higher neutralizing antibodies against all Omicron subvariants. This study provides evidence that BNT162b2 (BA.4/BA.5) and mRNA-1273 (BA.1) induce similar neutralization against Omicron subvariants, even when antigenically divergent from the circulating variant.

Severe acute respiratory syndrome coronavirus 2 (SARS-CoV-2), which causes Coronavirus Disease 2019 (COVID-19), has undergone continuous evolution leading to the emergence of novel variants of concern with high rates of breakthrough infections. Therefore, bivalent vaccines targeting the ancestral wild-type (D614G) spike and B.1.1.529 (Omicron) subvariants were approved for use[1–3]. In Canada, the mRNA-1273 BA.1 COVID-19 vaccine was approved September 1, 2022 while the BNT162b2 BA.4/5 COVID-19 vaccine was approved October 7, 2022.

[1]Division of Nephrology, Department of Medicine, Sunnybrook Health Sciences Centre, Temerty Faculty of Medicine, University of Toronto, Toronto, ON, Canada. [2]Division of Nephrology, Department of Medicine, University Health Network, Temerty Faculty of Medicine, University of Toronto, Toronto, ON, Canada. [3]Lunenfeld-Tanenbaum Research Institute at Mount Sinai Hospital, Sinai Health, Toronto, ON, Canada. [4]Division of Nephrology, Michael Garron Hospital, Toronto, ON, Canada. [5]Division of Infectious Diseases, Michael Garron Hospital, Toronto, ON, Canada. [6]Division of Experimental Medicine, Department of Medicine, University of British Columbia, Vancouver, BC, Canada. [7]British Columbia Provincial Renal Agency, Vancouver, BC, Canada. [8]Centre for Clinical Trial Support, Sunnybrook Research Institute, Toronto, ON, Canada. [9]Division of Nephrology, Department of Medicine, Unity Health Toronto, Temerty Faculty of Medicine, University of Toronto, Toronto, ON, Canada. [10]Division of Infectious Diseases, Department of Medicine, Sunnybrook Health Sciences Centre, Temerty Faculty of Medicine, University of Toronto, Toronto, ON, Canada. [11]Ontario Renal Network, Toronto, ON, Canada. [12]Department of Molecular Genetics, University of Toronto, Toronto, ON, Canada. ✉e-mail: michelle.hladunewich@sunnybrook.ca

**Table 1 | Baseline characteristics by bivalent vaccine type**

| Characteristic | No. (%) | | |
|---|---|---|---|
| | Total (*n* = 98) | mRNA-1273 (BA.1) (*n* = 72) | BNT162b2 (BA.4/ BA.5) (*n* = 26) |
| **Demographics** | | | |
| Age, median (IQR), y | 70 (61, 77) | 69 (61, 75) | 70 (62, 78) |
| Sex (Female) | 34 (35) | 21 (29) | 13 (50) |
| Race | | | |
| Asian | 26 (27) | 20 (28) | 6 (23) |
| Black | 7 (7) | 6 (8) | 1 (4) |
| Caucasian | 38 (39) | 33 (46) | 5 (19) |
| Other/unknown | 27 (28) | 13 (18) | 14 (54) |
| Patient type | | | |
| Dialysis | 83 (85) | 57 (79) | 26 (100) |
| Kidney transplant | 15 (15) | 15 (21) | 0 (0) |
| Cause of chronic kidney disease | | | |
| Diabetes | 29 (30) | 22 (31) | 7 (27) |
| Glomerulonephritis | 21 (21) | 13 (18) | 8 (31) |
| Hypertension | 15 (15) | 11 (15) | 4 (15) |
| Other | 33 (33) | 26 (36) | 7 (27) |
| Prior COVID-19 (RT-PCR or rapid antigen test) | 25 (26) | 19 (26) | 6 (23) |
| **Vaccine Type** | | | |
| Dose 1 | | | |
| ChAdOx1 | 3 (3.1) | 2 (2.8) | 1 (3.8) |
| mRNA-1273 | 36 (37) | 35 (49) | 1 (3.8) |
| BNT162b2 | 59 (60) | 35 (49) | 24 (92) |
| Dose 2 | | | |
| ChAdOx1 | 3 (3.1) | 2 (2.8) | 1 (3.8) |
| mRNA-1273 | 40 (41) | 39 (54) | 1 (3.8) |
| BNT162b2 | 55 (56) | 31 (43) | 24 (92) |
| Dose 3 | | | |
| mRNA-1273 | 56 (57) | 45 (62) | 11 (42) |
| BNT162b2 | 42 (43) | 27 (38) | 15 (58) |
| Dose 4 | | | |
| mRNA-1273 | 65 (66) | 56 (78) | 9 (35) |
| BNT162b2 | 26 (27) | 10 (14) | 16 (62) |
| mRNA-1273 Bivalent (BA.1) | 6 (6.1) | 6 (8.3) | 0 (0) |
| BNT162b2 Bivalent (BA.4/BA.5) | 1 (1.0) | 0 (0) | 1 (3.8) |
| Dose 5 | | | |
| mRNA-1273 Bivalent (BA.1) | 66 (73) | 66 (100) | 0 (0) |
| BNT162b2 Bivalent (BA.4/BA.5) | 24 (27) | 0 (0) | 24 (100) |
| Bivalent vaccine dose number | | | |
| Dose 4 | 8 (8.2) | 3 (4.1) | 5 (20) |
| Dose 5 | 90 (92) | 70 (96) | 20 (80) |
| Comorbidities | | | |
| Chronic obstructive pulmonary disease | 8 (8.2) | 5 (6.9) | 3 (12) |
| Coronary artery disease | 14 (14) | 7 (9.7) | 7 (27) |
| Congestive heart failure | 15 (15) | 10 (14) | 5 (19) |
| Cerebrovascular disease | 10 (10) | 7 (9.7) | 3 (12) |
| Hypertension | 80 (82) | 55 (76) | 25 (96) |

**Table 1 (continued) | Baseline characteristics by bivalent vaccine type**

| Characteristic | No. (%) | | |
|---|---|---|---|
| | Total (*n* = 98) | mRNA-1273 (BA.1) (*n* = 72) | BNT162b2 (BA.4/ BA.5) (*n* = 26) |
| Malignancy | 9 (9.7) | 7 (9.7) | 2 (9.5) |
| Peripheral vascular disease | 8 (8.2) | 6 (8.3) | 2 (7.7) |
| Immunosuppression | 17 (17) | 16 (22) | 1 (3.8) |
| Prednisone | 16 (16) | 15 (94) | 1 (100) |
| Mycophenolic acid | 12 (12) | 12 (75) | 0 (0) |
| Calcineurin inhibitor | 16 (16) | 16 (100) | 0 (0) |

Chronic kidney disease is a major risk factor for severe COVID-19[4] and those receiving hemodialysis are at increased risk for exposure to SARS-CoV-2[5]. Although patients with chronic kidney disease exhibit a robust antibody response to third-dose vaccination, new Omicron subvariants including BQ.1.1 and XBB.1.5 have immune-evasive potential[6,7]. Therefore, we evaluated the neutralizing antibody response to Omicron subvariants following bivalent COVID-19 vaccination in 98 patients receiving hemodialysis and kidney transplant recipients. Given that the mRNA-1273 bivalent vaccine targets BA.1 while BNT162b2 targets BA.4/5, we examined differences in neutralizing antibody levels against BA.1, BA.5, BQ.1.1, and XBB.1.5 induced by the different bivalent vaccines using a spiked-pseudotyped lentiviral neutralization assay.

## Results

Baseline characteristics of the 98 participants stratified by bivalent vaccine type were well balanced (Table 1). Median age was 70 years and 34% were female. The bivalent COVID-19 vaccine was the fifth overall dose in 92%, with 73% (72/98) receiving mRNA-1273 and 27% (26/98) receiving BNT162b2.

In samples taken prior to receipt of the bivalent vaccine, the median time from the most recent COVID-19 vaccine dose was 236 days (interquartile [IQR] 189–290). Following receipt of the bivalent vaccine dose, serum samples were taken at a median of 25 days (IQR 24–27). Among participants, 26% (25/98) had prior confirmed COVID-19, as determined by RT-PCR or rapid antigen testing, while 41% (40/98) had a positive anti-nucleocapsid antibody before bivalent vaccination. At 1 month follow-up, 37% (36/98) had a positive anti-nucleocapsid antibody with one new seroconversion while 5 individuals who were initially seropositive became seronegative. No clinical COVID-19 infections were reported during the study period (Supplementary Table 1).

Neutralizing antibodies increased a median of 4.5-fold (IQR 11.5) for wild-type, 8.8-fold (IQR 41.0) for BA.1, 7.8-fold (IQR 53) for BA.5, 8.5-fold for BQ.1.1 (IQR 66.2), and 8.1-fold (IQR 194) for XBB.1.5 following bivalent vaccination. The proportion of patients with detectable neutralization increased significantly from baseline to one-month post-vaccination for all subvariants ($p < 0.001$ for all): wild-type: 95% to 99%, BA.1: 76% to 93%, BA.5: 74% to 96%, BQ.1.1: 55% to 84%, XBB.1.5: 48% to 81% (Fig. 1; Supplementary Fig. 1 and Supplementary Table 2). Compared to the wild-type strain, median neutralizing antibody levels were attenuated by 7.3-fold for BA.1 (IQR 17.5), 8.3-fold for BA.5 (IQR 19.2), 45.8-fold for BQ.1.1 (IQR 170.3), and 48.2-fold (IQR 376.7) against XBB.1.5 (Supplementary Table 3).

While absolute neutralizing antibody levels were higher among those receiving the BNT162b2 vaccine, one month following bivalent vaccination, these differences were not statistically different by vaccine type for wild-type ($P = 0.48$), BA.1 ($P = 0.21$), BA.5 ($P = 0.069$), BQ.1.1

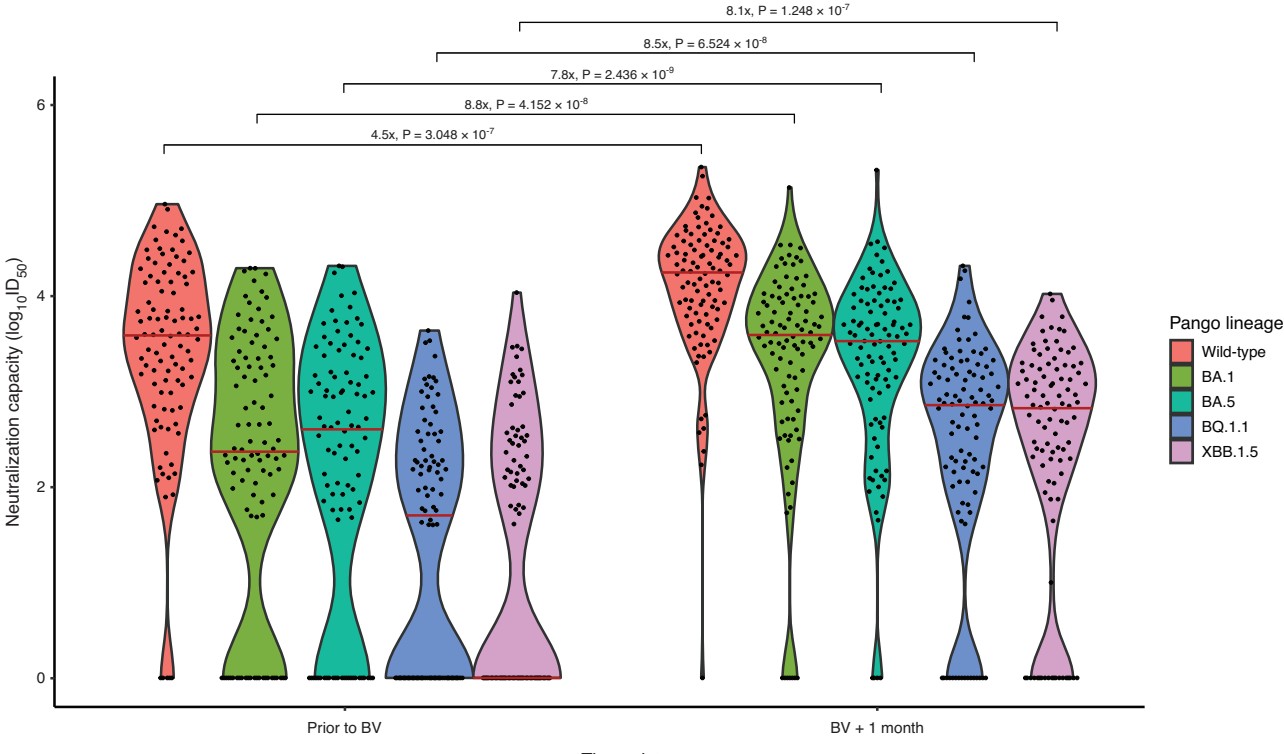

**Fig. 1 | Neutralizing capacity against SARS-CoV-2 Omicron subvariants prior to and 1 month following bivalent mRNA COVID-19 vaccination.** $Log_{10}$ $ID_{50}$ greater than 0 was considered detectable neutralization capacity. Dots represent individual serum samples collected ($n$ = 98 for each time point). Solid red line indicates median level. Fold change in neutralization capacity was 7.3-fold lower for BA.1, 8.3- fold lower for BA.5 and 45.8-fold lower for BQ.1.1 and 48.2-fold lower for XBB.1.5. in comparison to the wild-type (D614G) ancestral strain. Comparison prior to and following bivalent vaccination were evaluated using Wilcoxon signed-rank test with a two-sided $p$-value. No adjustments were made for multiple comparisons.

($P$ = 0.10), or XBB.1.5 ($P$ = 0.099) subvariants after adjustment for pre-specified covariates (Figs. 2a, 3 and Supplementary Table 4). The lack of difference between vaccine types was unchanged when the analysis was restricted to only hemodialysis participants: wild-type ($P$ = 0.50), BA.1 ($P$ = 0.25), BA.5 ($P$ = 0.098), BQ.1.1 ($P$ = 0.14), or XBB.1.5 ($P$ = 0.13) (Supplementary Fig. 2). Similarly, when we further adjusted for initial three dose vaccine types, no difference was observed between the BNT162b2 BA.4/5 and mRNA-1273 BA.1 bivalent vaccines: wild-type ($P$ = 0.26), BA.1 ($P$ = 0.49), BA.5 ($P$ = 0.23), BQ.1.1 ($P$ = 0.50), or XBB.1.5 ($P$ = 0.39) (Supplementary Table 5). There was no significant difference in neutralizing antibodies against any subvariant when comparing participants who had received the bivalent vaccine as the fourth dose ($n$ = 8) versus the fifth dose ($n$ = 90) (Fig. 4 and Supplementary Table 6).

We observed that participants with prior COVID-19, as determined by rapid antigen testing or RT-PCR, had higher absolute neutralizing antibody titers against all variants of concern following bivalent vaccination in comparison to those without prior COVID-19, although this was only statistically higher for BA.5 ($P$ = 0.023), BQ.1.1 ($P$ = 0.0017), and XBB.1.5 ($P$ = 0.049) (Supplementary Table 7).

When we determined prior COVID-19 infection instead by positive anti-nucleocapsid antibody rather than clinical infection to detect asymptomatic COVID-19, Omicron-specific neutralizing antibodies were significantly higher among those with a positive anti-nucleocapsid antibody against wild-type ($P$ = 0.047), BA.1 ($P$ = 0.0048), BA.5 ($P$ = 0.029), BQ.1.1 ($P$ = 0.018), and XBB.1.5 ($P$ = 0.014) (Fig. 2b).

One month following bivalent vaccination, kidney transplant recipients ($n$ = 15) had significantly lower absolute neutralizing antibody levels than hemodialysis patients ($n$ = 83) against BA.1 ($P$ = 0.044), BQ.1.1 ($P$ = 0.015), and XBB.1.5 ($P$ = 0.025), but not wild-type ($P$ = 0.42) or BA.5 ($P$ = 0.063) (Fig. 2c). Increases in neutralizing

antibody levels from baseline were significantly lower in kidney transplant recipients than hemodialysis for wild-type ($p$ < 0.001), BA.1 ($P$ = 0.008), BA.5 ($P$ = 0.010), BQ.1.1 ($P$ = 0.030) and XBB.1.5 ($P$ = 0.010) (Supplementary Table 8).

Following bivalent vaccination, binding IgG antibodies for anti-spike increased from 1.16 (IQR 0.77, 1.65) to 1.76 (IQR 1.52,1.86) 1-month post-vaccination and anti-RBD increased from 0.45 (IQR 0.18, 0.97) to 1.24 (IQR 0.80, 1.53) 1-month post-vaccination ($p$ < 0.0001 for both) in the full cohort (Supplementary Fig. 3).

## Discussion

Concerns have emerged regarding immune escape with newer Omicron subvariants[8,9]. Consistent with this, we observed a 46-fold decrease in viral neutralization against BQ.1.1 and 48-fold decrease against XBB.1.5 in comparison to wild-type. Consistent with other reports, we found that hybrid immunity enhanced neutralizing antibody response[10]. In addition, while BNT162b2 elicited higher absolute neutralizing antibodies in comparison to mRNA-1273 against all Omicron subvariants, these differences were not statistically significant after accounting for confounders including prior COVID-19 infection, number of vaccine doses, and hemodialysis versus kidney transplant recipients. Given that XBB.1.5 is presently the dominant circulating Omicron subvariant worldwide, this suggests that boosters targeting earlier subvariants may provide similar protection. Our results are consistent with a case series of dialysis patients, where bivalent vaccination increased anti-spike IgG 2.5 fold which correlated with BA.4/5 neutralization[11]. Our study findings conducted in a real-world setting are similar to the findings of a randomized controlled trial in 202 individuals, which compared the Pfizer/BioNTech BA.1 versus BA.4/5 vaccine and found similar neutralizing antibodies against BQ.1.1 and

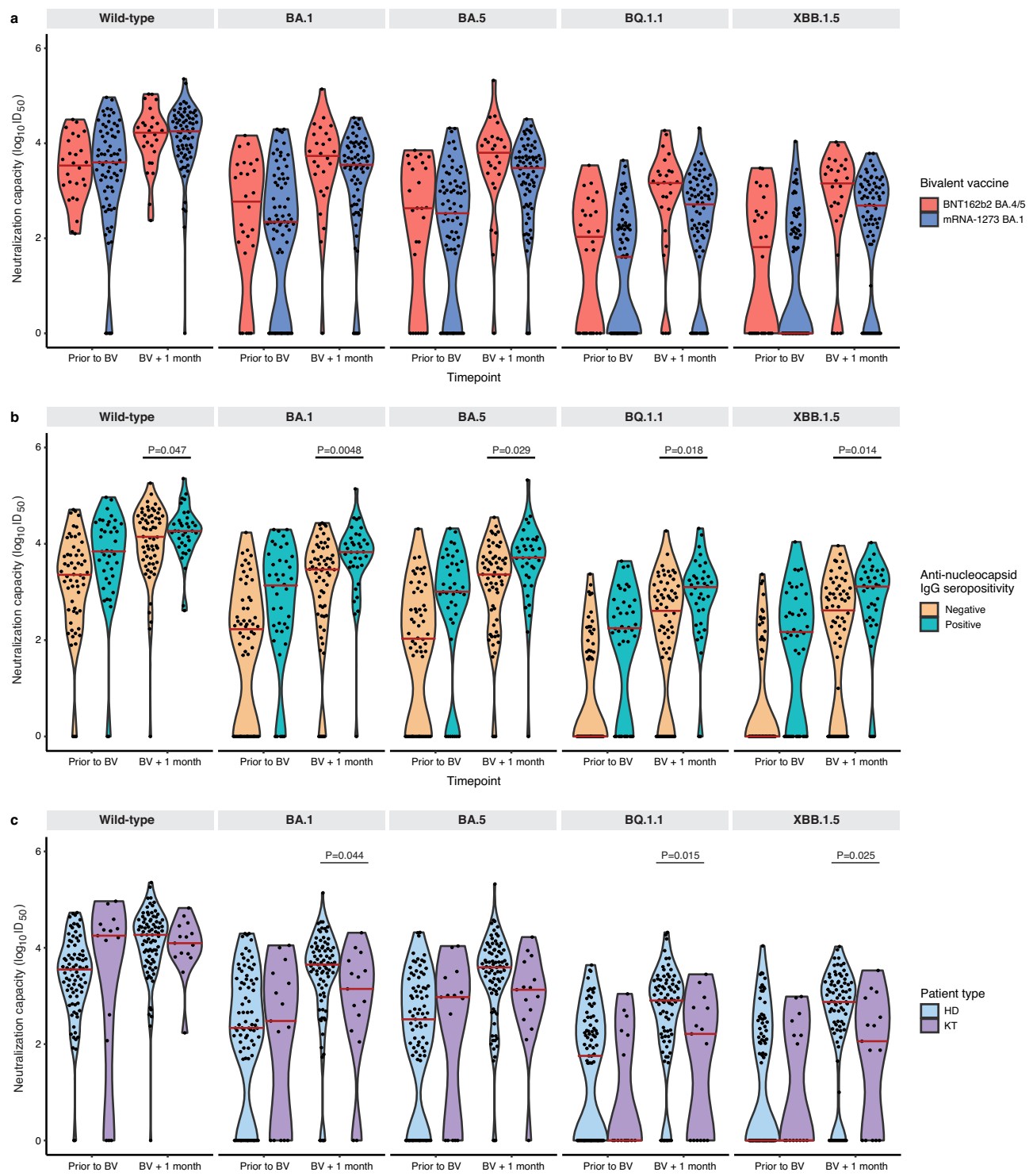

**Fig. 2 | Neutralizing antibodies against wild-type, BA.1, BA.5, BQ.1.1, and XBB.1.5 prior to and following bivalent vaccination. a** Stratified by vaccine type BNT162b2 BA.4/5 (n = 26) versus mRNA-1273 BA.1 (n = 72). Increases in neutralizing antibody levels were not significantly different by bivalent vaccine type for any subvariant after adjustment for anti-nucleocapsid positivity, hemodialysis versus kidney transplant recipient, and number of vaccine doses or (**b**) stratified by anti-nucleocapsid IgG seropositivity (n = 40) as a marker of prior COVID-19 infection versus seronegative (n = 58). Neutralizing antibody against Omicron subvariants were higher among those with anti-nucleocapsid seropositivity against wild-type (D614G) (P = 0.047), BA.1 (P = 0.0048), BA.5 (P = 0.029), BQ.1.1 (P = 0.018), and XBB.1.5 (P = 0.014) after adjusting for number of doses, vaccine type, patient type, and timepoint; or (**c**) stratified by patient type (hemodialysis n = 83) versus kidney transplant (n = 15). Solid red line indicates median level. Dots represent individual serum samples collected (n = 98 for each time point). Results were analysed using a linear mixed effects model, with a two-sided p-value. No adjustments were made for multiple comparisons.

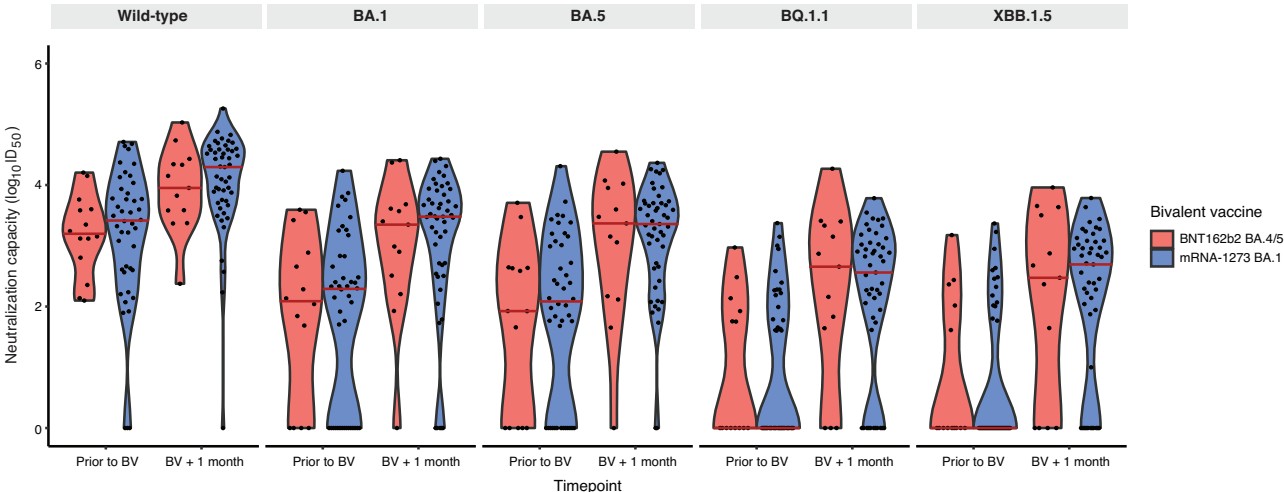

**Fig. 3 | Neutralizing antibodies against wild-type, BA.1, BA.5, BQ.1.1, and XBB.1.5 by bivalent vaccine type after exclusion of participants with a positive anti-nucleocapsid antibody.** Solid red line indicates median level. Dots represent individual serum samples collected ($n = 58$ prior to bivalent vaccination [BNT162b2 BA.4/5 $n = 14$; mRNA-1273 $n = 44$], $n = 52$ one-month post-vaccination [BNT162b2 BA.4/5 $n = 13$; mRNA-1273 $n = 39$].

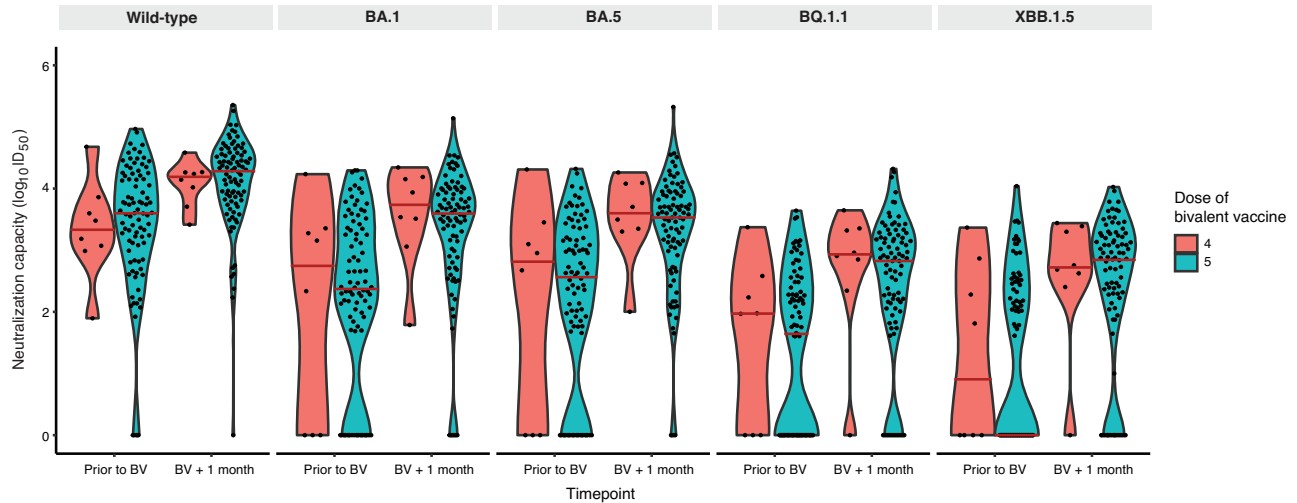

**Fig. 4 | Difference in neutralizing antibodies against wild-type, BA.1, BA.5, BQ.1.1, and XBB.1.5 by total number of COVID-19 vaccine doses.** There was no significant difference in neutralizing antibodies against any subvariant in those receiving 4 ($n = 8$) versus 5 doses ($n = 90$). Solid red line indicates median level. Dots represent individual serum samples collected ($n = 98$ for each timepoint).

XBB.1.5 with either bivalent vaccine[12]. However, this randomized trial differed from our study as it only included healthy individuals and did not evaluate the mRNA-1273 BA.1 vaccine.

Other studies have not found a higher peak neutralizing antibody response with bivalent vaccines in comparison to monovalent vaccines, and suggested a role for immunologic imprinting wherein prior antigenic exposures prime B cell memory and limit the development of memory B cells and neutralizing antibodies against newer subvariants[8,13]. Kidney transplant recipients generally had both lower increases and absolute lower neutralizing antibodies levels post-vaccination, which is unsurprising given that transplant recipients are known to have attenuated humoral response to COVID-19 vaccination[14]. The number of prior vaccine doses (4 versus 5) did not meaningfully influence neutralizing antibody levels both prior to and following bivalent vaccination.

Strengths of this study include that it is the first to compare neutralizing antibody response against current circulating Omicron subvariants with bivalent vaccines targeting BA.1 versus BA.4/5. We had anti-nucleocapsid serology and clinical information available on all participants, allowing us to account for multiple factors including prior COVID-19 infection in our analysis. This study also focuses on a vulnerable patient population as both kidney transplant recipients and individuals receiving hemodialysis have lower humoral responses to COVID-19 vaccination and are at higher risk for severe COVID-19 outcomes[15,16]. Therefore, characterizing the dynamics of humoral response to bivalent vaccination will help to inform future vaccination strategies in vulnerable populations, including the timing of additional boosters. Similarly, a recent study in patients with hematologic malignancy demonstrated the utility of serologic testing in immunocompromised populations for predicting future risk of COVID-19 infection[17].

This study does have limitations. Due to the observational nature of this study, there may have been residual confounding. Individuals who chose the mRNA-1273 BA.1 vaccine, which was available earlier in Canada, were more likely to have received prior mRNA-1273 vaccination in comparison to those who received the BNT162b2 BA.4/5 vaccine. However, in a sensitivity analysis wherein we adjusted for the initial three-dose vaccine types, the underlying conclusions were unchanged. Due to temporal differences in mRNA-1273 BA.1 versus

BNT162b2 BA.4/5 vaccine availability, it is possible that differences in COVID-19 infections could have occurred based upon epidemiologic and participant factors. However, only one individual contracted COVID-19 (as determined by anti-nucleocapsid seroconversion) during follow-up. As such, COVID-19 infections were unlikely to have affected the underlying study conclusions. We did not evaluate cellular immunity, although prior studies have suggested that spike-specific CD8+ and CD4 + T-cell response increased only modestly after bivalent mRNA boosting[18]. In addition, while neutralizing antibody levels correlate with vaccine effectiveness[19,20], the clinical implications of these findings on preventing COVID-19 infection require confirmation at a population-level, as there is evidence that bivalent vaccine effectiveness declines after one month[21]. Finally, we were not able to evaluate the mRNA-1273 BA.4/5 bivalent vaccine as it was not approved in our jurisdiction during the study period.

In conclusion, we found that bivalent vaccines increased neutralizing antibodies against Omicron subvariants, although neutralizing antibody titers were significantly attenuated against BQ.1.1 and XBB.1.5. We found that both BNT162b2 and mRNA-1273 bivalent vaccines elicited a similar neutralizing antibody response against Omicron subvariants including BQ.1.1 and XBB.1.5, suggesting that variant modified bivalent vaccines may confer protection against emerging circulating Omicron subvariants despite antigenic divergence.

## Methods

This study protocol was approved by the respective Institutional Review Boards at Sunnybrook Health Sciences Centre and Unity Health Network (CTO #3604) as well as Michael Garron Hospital (REB # 856-2201-Inf-066). All procedures were in accordance with the ethical standards of the 1964 Declaration of Helsinki. This was a prospective observational cohort study involving individuals age ≥18 receiving hemodialysis or with a kidney transplant receiving the mRNA-1273 bivalent vaccine (Original and Omicron BA.1) or BNT162b2 bivalent vaccine (Original and Omicron BA.4/BA.5). The study was conducted between July 25 and November 30, 2022. The protocol for this study has been published previously[22]. Only participants unable to provide informed consent due to cognitive impairment or a language barrier if a translator was unavailable were excluded. Demographics, vaccination status, comorbidities, and medications were obtained from electronic patient records. Serum samples were taken prior to and one month following bivalent vaccination to evaluate neutralizing antibody levels against Omicron subvariants. Serum samples were collected in a serum separator tube and allowed to clot for 30 minutes and centrifuged for 10 minutes at $3000 \times g$-force at room temperature. Levels of binding IgG antibodies to each of the antigens (produced by the National Research Council of Canada) were normalized to reference standards included on each plate and expressed as relative ratios or World Health Organization International Standard units (BAU/mL).

### Spiked-Pseudotyped lentiviral neutralization

The spiked-pseudotyped lentiviral neutralization assay was performed in HEK293T-ACE2/TMPRSS2 cells[14]. Lentiviral virus-like particles were generated from co-transfection of the viral packaging (psPAX2, Addgene, #12260), the ZsGreen and luciferase reporter construct (pHAGE-CMV-Luc2-IRES-ZsGreen-W, provided by Jesse Bloom), and the spike protein constructs: wild-type (D614G), Omicron BA.1, BA.5, BQ.1.1, XBB.1.5 subvariants available at http://nbcc.lunenfeld.ca/resources), into HEK293TN cells (System Biosciences [LV900A-1]). To construct HDM_XBB.1.5, a human codon-optimized cDNA encoding the XBB.1 spike variant (Twist Bioscience) was first inserted into the mammalian expression plasmid HDM (a gift from Jesse Bloom) and modified to include the F486P mutation starting from the cDNA fragment encoding the XBB.1.5 RBD. Viral supernatants were harvested, clarified, and filtered through 0.45-μm filters prior to storage at −80 °C. A viral titer assay was performed by infecting HEK293T-ACE2/TMPRSS2 cells, followed by a luciferase assay to determine the relative luciferase unit (1:10 to 1:250 dilution of virus stock, depending on the virus titers of each variant). For the neutralization assay, diluted patient sera samples (1:22.5) were prepared and serially diluted by 3-fold over 7 dilutions, followed by incubation with diluted pseudo-virus for 1 h at 37 °C prior to addition to HEK293T-ACE2/TMPRSS2 cells. Cells were lysed 48-hours post infection using the Bright-Glo Luciferase Assay System (Promega), and the luminescence signals were detected using a PerkinElmer EnVision instrument. HEK293TN and HEK293T-ACE2/TMPRSS2 cells are maintained at 85% confluency for no more than 25 passages. The inhibitory dilution with 50% virus neutralization (ID50) was calculated in GraphPad Prism version 9.5.0 using a nonlinear regression algorithm (log[inhibitor] versus normalized response - variable slope). In patients with an absence of 50% neutralization with undiluted serum, a $\log_{10}$ $ID_{50}$ of zero was considered the threshold for detectable viral neutralization.

### SARS-CoV-2 binding IgG antibodies

Binding IgG antibodies against the ancestral (D614G) full-length spike protein (anti-spike), its receptor binding domain (anti-RBD) (Identifier: VHH72-hFc1X7), and anti-nucleocapsid antibodies (HC2003; Source: Genscript; Identifier: Cat#A02039) were validated and measured on an automated enzyme-linked immunosorbent assay platform[23]. Binding antibody levels were reported as relative ratios to a synthetic standard included as a calibration curve on each assay plate. Thresholds for binding antibody seroconversion were determined by aggregating data from negative controls and calculating the mean +3 standard deviations (SD) and were measured at a 1:160 (0.0625 μL/well) dilution. Negative controls were obtained from pre-COVID-19 pandemic sera, blanks, and commercially purified IgG. The threshold for seropositivity was obtained from the mean of the log distributions of the controls which was found to result in a specificity for RBD of 100%, a sensitivity of 89%, while spike had a sensitivity of 99% and specificity of 94%. Seroconversion thresholds were 0.186, 0.19, and 0.164 for anti-RBD, anti-spike, and anti-nucleocapsid antibodies, respectively.

### Statistical analysis

Descriptive statistics were used to describe baseline characteristics. Neutralizing antibodies were analyzed through linear mixed-effects models with random intercepts and fixed covariates adjusting for vaccine time point, bivalent vaccine type, number of COVID-19 vaccine doses (four versus five), kidney transplant recipients versus dialysis patients, and anti-nucleocapsid antibody status. In a sensitivity analysis, we additionally adjusted for initial two-dose vaccine and third-dose vaccine type to account for differences in immunogenicity between monovalent COVID-19 vaccines. With two-sided testing, $P = 0.05$ was considered statistically significant. We performed all analyses using R version 4.0.5 (R Project for Statistical Computing).

### Reporting summary

Further information on research design is available in the Nature Portfolio Reporting Summary linked to this article.

## Data availability

The serologic data generated in this study are openly available through the publicly accessible Borealis repository (https://doi.org/10.5683/SP3/9XUY6O). The patient-specific clinical data are available under restricted access through the publicly accessible CITF Databank. The source data generated in this study for figures are provided in the Supplementary Information/Source Data file provided with this paper.

## Code availability

The code for this study is publicly available at https://github.com/covidckd/bivalent.git.

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

## Acknowledgements

This study is supported by the COVID-19 Immunity Task Force grant 2122-HQ-000071 (A.L., M.J.O., and M.H.), which is funded by the Government of Canada. The Canadian Institutes of Health Research supports lentivirus assay development through CoVaRR-Net (Coronavirus Variants Rapid Response Network) (A.C.G.). The equipment used is housed in the Network Biology Collaborative Centre at the Lunenfeld-Tanenbaum Research Institute, a facility supported by Canada Foundation for Innovation funding (grant CFI 33474), by the Ontario Government, Genome Canada, and Ontario Genomics (OGI-139) (A.C.G.). K.Y. is supported by the University of Toronto Department of Medicine Eliot Phillipson Clinician Scientist Training Program, Banting and Best Diabetes Centre Postdoctoral Fellowship, and KRESCENT Postdoctoral Fellowship. The KRESCENT program is co-sponsored by the Kidney Foundation of Canada, the Canadian Society of Nephrology, and the Canadian Institutes of Health Research. D.A.Y. is supported by a Canada Research Chair, Tier 2, in Fibrotic Injury. A.C.G. is supported by the Canadian Institutes of Health Research (FDN 143301) and a Canada Research Chair, Tier 1, in Functional Proteomics and is a pillar lead for CoVaRR-Net. The antibody measurements and initial data analysis were performed by Bhavisha Rathod, Geneviève Mailhot, Mahya Fazel-Zarandi, Jenny Wang, Kento Abe and Adrian Pasculescu. We would like to thank Gail Klein and Mandana Rahimi for valuable assistance in coordinating this study.

## Author contributions

K.Y., Q.H., A.C.G. and M.H. designed the study. A.K., F.Q., Q.H. performed the data extraction, curation and analysis. K.Y., A.K., F.Q., Q.H. and Y.J. performed the data analysis. K.Y. and M.H. wrote the first draft of the manuscript. K.Y., A.K., F.Q., M.D.B., T.R.T., Q.H., M.D., C.K., O.E., A.L., Y.J., W.R.H., D.A.Y., J.P., C.T.C., J.A.L., M.J.O., K.C., A.C.G., M.A.H. contributed to the writing as well as reviewed and approved the final report.

## Competing interests

A.L. reports being a scientific advisor to, or member of, AstraZeneca, Bayer, Boehringer-Ingelheim, Canadian Journal of Kidney Health and Disease, Canadian Institutes of Health Research, Certa, Chinook Therapeutics, Johnson and Johnson, Kidney Foundation of Canada, National Institutes of Health (NIH), National Institute of Diabetes and Digestive and Kidney Diseases (NIDDK), Otsuka, Reata, Retrophin, and The George Institute; receiving research funding from AstraZeneca, Boehringer-Ingelheim, Canadian Institute of Health Research, Janssen, Johnson and Johnson, Kidney Foundation of Canada, Merck, NIDDK, NIH, Ortho Biotech, Otsuka, and Oxford Clinical Trials; and having consultancy agreements with Amgen, AstraZeneca, Bayer, Boehringer-Ingelheim, Johnson and Johnson/Jansen, Reata, and Retrophin (unrelated to the submitted work). D.Y. reports being a scientific co-founder and consultant for Fibrocor Therapeutics; receiving speaking honoraria and/or consultancy fees from AstraZeneca, GlaxoSmithKline, and Vivace Therapeutics (unrelated to the submitted work). J.P. reported receiving speaking honoraria and consultancy fees from Baxter Healthcare; grants from Agency for Healthcare Research and Quality grant support; speaking honoraria from Fresenius Medical Care, AstraZeneca, Davita Healthcare, and US Renal Care; and consultancy fees from LiberDi Dialysis (unrelated to the submitted work). J.L. has received payment for expert testimony upon request of hospitals of the Ontario Hospital Association, Ministry of Attorney General of Ontario, and Seneca College (unrelated to the submitted work). A.C.G. has received research funds from a research contract with Providence Therapeutics Holdings, Inc for other projects, participated in the COVID-19 Immunity Task Force Immune Science and Testing working party, chaired the CIHR Institute of Genetics Advisory Board, and chairs the SAB of the National Research Council of Canada Human Health Therapeutics Board. M.O. and M.H are contracted Medical Leads at the Ontario Renal Network, Ontario Health. Matthew Oliver is owner of Oliver Medical Management Inc., which licenses Dialysis Management Analysis and Reporting System software. He has received honoraria for speaking from Baxter Healthcare (unrelated to the submitted work). M.H. reports receiving grants from Pfizer for a study in focal segmental glomerulosclerosis, Ionis, Calliditas and

Chinook for studies in Immunoglobulin A nephropathy, and Roche for a preeclampsia study (unrelated to the submitted work). No other competing interests were declared.
