## [Peer Review File · Nature Communications]

Omicron variant Neutralizing Antibodies Following BNT162b2 BA.4/5 versus mRNA-1273 BA.1 Bivalent Vaccination in Patients with End-Stage Kidney DiseaseReviewers' Comments:

Reviewer #1:

Remarks to the Author:

Yau et al analysed the neutralising ability of sera against Omicron BA.1, BA.5, BQ.1.1, and XBB.1.5 following a booster bivalent vaccine dose (BNT162b2 (BA.4/BA.5) or mRNA-1273 (BA.1)) in 98 individuals receiving dialysis or with a kidney transplant. The authors used a spiked-pseudotyped lentiviral neutralization assay and they found that the neutralisation ability was enhanced by both types of booster vaccine dose even though it still remained lower against the range of Omicron variants compared to the ancestral strain. There was not significant difference between the type of bivalent vaccines even though individuals who received the BNT162b2 vaccine had slightly higher neutralising responses against Omicron variants one month following vaccination compared to mRNA-1273 recipients. In addition, there was no significant difference in neutralising responses in individuals who had received the dose as 4th or 5th dose. Finally, people with positive anti-N antibody before vaccination had significantly higher neutralising responses against the ancestral strain and Omicron variants before and after vaccination compared to individuals with anti-N negative at the same timepoint.

Even though this paper is limited to neutralising responses, it remains relevant. It is essential to understand antibody responses against new Omicron variants induced by the bivalent vaccines. The paper is well written, the figures and the take-home messages are clear. Please find below a few comments.

Comments:

- 1) Could the authors show significant differences in the graphs ?
- 2) Extended data figure 2: after exclusion of individuals with positive anti-nucleocapsid antibody, it looks the trend showing higher neutralising responses after BNT162b2 BA4/5 is lost. Does it mean that individuals with positive anti-nucleocapsid antibody had recently been infected and probably with BA4 or 5 ? Which could explain why BNT162b2 BA4/5 induced slightly stronger neutralising responses especially in recently infected individuals ?!
- 3) Extended data figure 4: I presume the authors used ancestral SARS-CoV-2 spike and RBD as antigens. Did the authors also analyse IgG binding to Omicron variant spike or RBD ?

Minor comments:

- 1) "Other studies have found not found a higher peak neutralizing antibody response with bivalent vaccines in comparison to monovalent vaccines..."
I presume it is "have not found" ? Please clarify.
- 2) "The spiked-pseudotyped lentiviral neutralization assay was performed in HEK293T-ACE2/TMPRSS2 cells was performed as described previously..."
"Was performed" has been repeated twice.
- 3) "Thresholds for binding antibody seroconversion were determined by aggregating data from negative controls and calculating the mean +3 standard deviations (SD)..."
I presume negative controls are pre-pandemic sera. Could the authors clarify it in the text ?

Reviewer #2:

Remarks to the Author:

In this manuscript, Yau and colleagues describe the results of a prospective, observational study comparing the short-term cross-Omicron immunogenicity of a Pfizer BA4/5 vs a Moderna BA1 bivalent COVID-19 vaccine in patients with renal disease on dialysis or post-transplant in Canada. The investigators found that both bivalent vaccines boosted cross-Omicron responses above baseline, but that there was no significant difference between the two different vaccines. Bivalent boosting in participants with evidence of prior SARS-CoV-2 infection led to the highest titers. Transplant patients

did not respond well to either boosting regimen, particularly against recent, immune evasive XBB variants.

The strengths of this study are as follows:

- Takes advantage of the fact that in Canada, both a BA.1 and a BA.4/5 bivalent vaccine were authorized for use and can therefore be compared. (In the US, the BA.1 vaccines were not authorized).
- Prospective study
- Includes a novel population of patients with chronic kidney disease including a smaller cohort of renal transplant recipients
- Uses an extended panel of pseudoneutralization constructs with data for the most recent Omicron subvariants including XBB.
- Addresses immunogenicity in a relatively older cohort (median age 69 years)
- Controls for prior infection as measured by anti-NC seropositivity.

Weaknesses

- Observational
- Short follow up of 30 days
- Study population is not necessarily generalizable
- There is another randomized controlled trial looking at bivalent BA4/5 and bivalent BA1 immune responses with similar findings (Branche AR Clin Infect Dis. 2023;ciad209. doi:10.1093/cid/ciad209). This diminishes the impact of the study submitted here. The RCT was in a different population (not CKD patients) and not in a real-world setting, and didn't include the Pfizer vaccine. So there is a difference, but it needs to be noted
- The study is comparing Pfizer BA4/5 to Moderna BA1, and there are differences in the mrna dose and platform immunogenicity that might be confounding results
- The study does not control for the fact that the Moderna BA1 population may be qualitatively different than the BA4/5 population. For example, Moderna BA1 was authorized in Canada a month before Pfizer BA4/5. The extended data shows that all the people who waited for Pfizer had gotten all their previous doses with Pfizer, whereas the Moderna group had more heterogenous prior vaccines. This can influence immunogenicity. Then there is the issue of early vs. late adopters (Moderna adopters were early, Pfizer bivalents were late); this may track with the frequency and nature of prior SARS-CoV-2 exposures and the subsequent character of immune responses following bivalent boost. These limitations should be discussed.
- In Figure 2, there appear to be a lot of NC seroconversions between baseline and post-bivalent x 1 month. The methods say that baseline samples were collected 'prior to' bivalent dosing, and that the study started in July 2022 – so when were the baseline samples obtained, and was it the same for everyone? If there were collected right before bivalent dosing (say that morning), then the data would suggest that a lot of people had breakthrough infections in the 30 days post-vaccine. This definitely happens. But that could really diminish the ability to detect differences between the groups - because if both groups had lots of BA5 exposure then that might swamp the impact of the BA4/5 or BA1

vaccine.

- Transplant patients have a very different host immune status compared to hemodialysis patients, depending on the level of immunosuppression present. This is born out in the Figure 2c data which shows how poorly the transplant patients responded to vaccine. I'm not sure it makes sense to lump them together when looking at all the other analyses. Especially because I think the transplant patients were exclusively in the Moderna group as per the extended data.

Summary

This paper takes advantage of a unique opportunity to compare BA4/5 and BA1 bivalent vaccines in a real-world setting, with a special emphasis on patients with chronic kidney disease who are either receiving hemodialysis or post-transplant. The study adds to the literature showing that multi-valent vaccines may not need to exactly match a currently circulating strain in order to provide increased breadth. This has important implications for how governments and industry will invest in new COVID-19 vaccine updates. The paper's impact is diminished by an existing RCT that also compares BA1 vs. BA4/5 bivalent vaccines. Nevertheless, it shows that these results are borne out in a real world setting and in a population with high risk for progression to severe disease in the setting of breakthrough infection.

The paper would benefit from highlighting the special population being studied, such as in the title. There should be a greater discussion of the limitations inherent to an observational study, with a particular emphasis on more subtle ways that comparative groups can be different.

Michelle Hladunewich MD, MSc, FRCPC
William Sibbald Chair for the Physician-in-Chief
Professor of Medicine, University of Toronto
Physician-in-Chief, Department of Medicine, Sunnybrook Health Sciences Centre
2075 Bayview Avenue, Room D4-74, Toronto, Ontario, Canada M4N 3M5
Tel: 416-480-4592 Fax: 416-480-6191 e-mail: michelle.hladunewich@sunnybrook.ca

Response to Reviewer Comments

Reviewer #1

Yau et al analysed the neutralising ability of sera against Omicron BA.1, BA.5, BQ.1.1, and XBB.1.5 following a booster bivalent vaccine dose (BNT162b2 (BA.4/BA.5) or mRNA-1273 (BA.1)) in 98 individuals receiving dialysis or with a kidney transplant. The authors used a spiked-pseudotyped lentiviral neutralization assay and they found that the neutralisation ability was enhanced by both types of booster vaccine dose even though it still remained lower against the range of Omicron variants compared to the ancestral strain. There was not significant difference between the type of bivalent vaccines even though individuals who received the BNT162b2 vaccine had slightly higher neutralising responses against Omicron variants one month following vaccination compared to mRNA-1273 recipients. In addition, there was no significant difference in neutralising responses in individuals who had received the dose as 4th or 5th dose. Finally, people with positive anti-N antibody before vaccination had significantly higher neutralising responses against the ancestral strain and Omicron variants before and after vaccination compared to individuals with anti-N negative at the same timepoint.

Even though this paper is limited to neutralising responses, it remains relevant. It is essential to understand antibody responses against new Omicron variants induced by the bivalent vaccines. The paper is well written, the figures and the take-home messages are clear. Please find below a few comments.

Comments:

1) Could the authors show significant differences in the graphs?

Thank you, we have now shown the significant differences as p-values in **Figure 1 and 2** as requested as shown below for ease of presentation.

Fig. 1 | Neutralizing capacity against SARS-CoV-2 Omicron subvariants prior to and 1 month following bivalent mRNA COVID-19 vaccination. Log₁₀ ID₅₀ greater than 0 was considered detectable neutralization capacity. Dots represent individual serum samples collected (n=98 for each time point). Solid red line indicates median level. Fold change in neutralization capacity was 7.3-fold lower for BA.1, 8.3-fold lower for BA.5 and 45.8-fold lower for BQ.1.1 and 48.2-fold lower for XBB.1.5. in comparison to the wild-type (D614G) ancestral strain.

Fig. 2 | Neutralizing antibodies against BA.1, BA.5, BQ.1.1, and XBB.1.5 prior to and following bivalent vaccination a) stratified by vaccine type BNT162b2 BA.4/5 (n=26) versus mRNA-1273 BA.1 (n=72). Increases in neutralizing antibody levels were not significantly different by bivalent vaccine type for any subvariant after adjustment for anti-nucleocapsid positivity, patient type, and number of vaccine doses or b) stratified by anti-nucleocapsid IgG seropositivity (n=40) as a marker of prior COVID-19 infection versus seronegative (n=58). Neutralizing antibody against Omicron subvariants were higher among those with anti-nucleocapsid seropositivity against wild-type (D614G) (P=0.047), BA.1 (P=0.0048), BA.5 (P=0.029), BQ.1.1 (P=0.018), and XBB.1.5 (P=0.014) after adjusting for number of doses, vaccine type, patient type, and timepoint; or c) stratified by patient type (hemodialysis n=83) versus kidney transplant (n=15). Solid red line indicates median level. Dots represent individual serum samples collected (n=98 for each time point).

2) Extended data figure 2: after exclusion of individuals with positive anti-nucleocapsid antibody, it looks the trend showing higher neutralising responses after BNT162b2 BA4/5 is lost. Does it mean that individuals with positive anti-nucleocapsid antibody had recently been infected and probably with BA4 or 5 ? Which could explain why BNT162b2 BA4/5 induced slightly stronger neutralising responses especially in recently infected individuals?

We included **Extended Data Figure 2** in order to demonstrate the degree to which prior COVID-19 infection (as determined by anti-nucleocapsid seropositivity) increases SARS-CoV-2 neutralizing antibody levels. After adjusting for anti-nucleocapsid seropositivity, this eliminated any difference in humoral response between the two bivalent vaccines evaluated in this study (BNT162b2 BA.4/5 vs. mRNA-1273 BA.1).

However, the positive anti-nucleocapsid seropositivity does not necessarily imply new infection here (particularly with BA.4/5 variants of concern), as these infections could have occurred at any point in time prior to the receipt of the bivalent vaccine.

New COVID-19 infections would likely have been BA.5 or XBB.1.5 based upon the epidemiology at the time. However, no participants reported COVID-19 during follow-up. In addition, we observed 92/98 (94%) concordance between anti-nucleocapsid status prior to bivalent vaccination and at 1 month follow-up. Among the 6 discordant pairs (i.e., a difference in anti-nucleocapsid status between baseline and follow-up), 5 out of 6 had a positive anti-nucleocapsid prior to bivalent vaccination with their anti-nucleocapsid value decreasing below the threshold for positivity during 1 month follow-up. These individuals were counted as having a positive anti-nucleocapsid, given that this indicated prior COVID-19 infection at baseline. Only one individual had new anti-nucleocapsid seroconversion during follow-up. To clarify this, we have added an additional **Extended Data Table 2** as follows:

Extended Data Table 2: Anti-nucleocapsid IgG seropositivity and seroconversion prior to and 1 month following bivalent vaccination.

Timepoint	Anti-Nucleocapsid IgG Seropositivity
Pre-Bivalent	40/98 (41%)
Bivalent Vaccine + 1 month	36/98 (37%)

At follow-up 1 seroconversion for anti-nucleocapsid IgG occurred and 5 participants initially seropositive for anti-nucleocapsid IgG became seronegative.

We have also clarified this in manuscript results as follows:

“Among participants, 26% (25/98) had prior confirmed COVID-19, as determined by RT-PCR or rapid antigen testing, while 41% (40/98) had a positive anti-nucleocapsid antibody before bivalent vaccination. At 1 month follow-up, 37% (36/98) had a positive anti-nucleocapsid antibody with one new seroconversion and 5 individuals who were initially seropositive became seronegative. No clinical COVID-19 infections were reported during the study period.”

3) Extended data figure 4: I presume the authors used ancestral SARS-CoV-2 spike and RBD as antigens. Did the authors also analyse IgG binding to Omicron variant spike or RBD?

This is correct, for the IgG antibody results shown in **Extended Data Figure 4**, we used the ancestral SARS-CoV-2 spike and RBD as antigens. We have clarified in our methods that the ancestral strain for was used for the binding IgG antibodies as follows:

“Binding IgG antibodies against the ancestral (D614G) full-length spike protein (anti-spike), its receptor binding domain (anti-RBD), and anti-nucleocapsid antibodies were measured on an automated enzyme-linked immunosorbent assay platform described previously.”

We did not examine Omicron variant spike or RBD IgG in the binding antibody ELISA analysis as the binding antibody measurements were intended to be complementary to the neutralizing antibody measures which better correlate with severe COVID-19 outcomes.

Notably, we have previously demonstrated correlation between ancestral SARS-CoV-2 RBD IgG binding antibodies and neutralizing antibody measures as shown in the Figure below. Here we showed that anti-RBD binding antibodies had high correlation with neutralizing antibodies against wild-type ($\rho=0.92$, $p<2.2E-16$), BA.1 ($\rho=0.86$, $p<2.2E-16$), BA.5 ($\rho=0.84$, $p<2.2E-16$), BQ.1.1 ($\rho=0.78$, $p<2.2E-16$), XBB.1.5 ($\rho=0.84$, $p<2.2E-16$) although the correlations were lowest for BA.5, BQ.1.1 and XBB.1.5.

We do note that at relative ratios above 1.2 the correlation between anti-RBD and neutralizing antibodies may be affected due to signal saturation of the anti-RBD. This is because the linear range of the IgG binding antibody assay is between a relative ratio to a synthetic standard of 0.031 (lower limit) to 1.2 (upper limit) for anti-RBD. Above the upper limit of 1.2, a further increase in antibody levels is no longer linear to the detection signal which indicates the signal is

saturated at that dilution. Above the linear range, we cannot accurately measure the antibody levels (we report as >1.2). Assigning these samples, the maximal measurable value leads to a compression of the data at the upper end which may mask biological differences.

Supplemental Figure | Correlation of neutralizing antibodies against wild-type SARS-CoV-2, Omicron (B.1.1.529) subvariants with SARS-CoV-2 anti-RBD IgG binding antibodies measured by ELISA at the 1:2560 (0.0039 μ L/well) dilution.

Minor comments:

1) *“Other studies have found not found a higher peak neutralizing antibody response with bivalent vaccines in comparison to monovalent vaccines” I presume it is “have not found” ? Please clarify.*

Thank you, we have corrected this as “have not found.”

2) *“The spiked-pseudotyped lentiviral neutralization assay was performed in HEK293T-ACE2/TMPRSS2 cells was performed as described previously” “Was performed” has been repeated twice.*

We have removed the duplicated “was performed” as noted.

3) *“Thresholds for binding antibody seroconversion were determined by aggregating data from negative controls and calculating the mean +3 standard deviations (SD)” I presume negative controls are pre-pandemic sera. Could the authors clarify it in the text?*

The negative controls were blanks, pre-COVID-19 negative sera, and commercially purified IgG. We have clarified in the manuscript as follows:

“Negative controls were obtained from pre-COVID-19 pandemic sera, blanks, and commercially purified IgG. The threshold for seropositivity was obtained from the mean of the log distributions of the controls which was found to result in a specificity for RBD of 100%, a sensitivity of 89%, while spike had a sensitivity of 99% and specificity of 94%.”

The development of the assays in this study have previously been described (Colwill *et al.* Clinical and Translation Immunology 2022) which we have shown an excerpt of below:

“We monitored the negative controls (blanks, pre-COVID-19 negative sera, commercially purified IgG, n = 1320 for spike and 1248 for N and RBD) used in 23 experiments conducted over 4 months. We calculated a threshold of 3 SDs from the mean of the log distribution of these controls and compared it to the thresholds established by ROC analysis. The thresholds were very similar for N and spike, but the 3 SD range was more stringent for the RBD, decreasing the likelihood of false positive calls. We therefore adopted a threshold of 3 SDs from the mean of these controls for each antigen and recalculated our performance characteristics. At this threshold, the specificity for RBD increased to 100% with a slight decrease in sensitivity to 89%, spike retained the same sensitivity and specificity (99% and 94%, respectively), whereas N’s sensitivity decreased from 81% to 79% with 99% specificity. Our final determination of sample positivity in seroprevalence settings therefore requires that it exceeds the thresholds of at least 2 out of 3 antigens.”

A Figure from this prior work shows how the threshold was determined below:

Figure 2. Development of high-throughput ELISAs for plasma or serum. **(a)** Known negative (pre-COVID-19) and positive (confirmed convalescent) samples (0.0625 $\mu\text{L}/\text{well}$) were tested in an automated antibody detection ELISA in two separate replicates 7 weeks apart. Spearman correlations are noted. **(b)** Density distributions of negative samples were plotted for each antigen. The black lines represent the mean of the negative distribution (dotted) and three SDs from the mean (solid; the relative ratio is indicated). The blue line represents the thresholds established by ROC analysis. **(c)** Comparison of the antigens with a set of known negative and positive samples at 0.0625 $\mu\text{L}/\text{well}$. Spearman correlations are shown. For both **a** and **c**, dashed lines represent the thresholds as defined by the 3-SD negative distribution shown in **b** and listed in Table 1.

Reviewer #2

In this manuscript, Yau and colleagues describe the results of a prospective, observational study comparing the short-term cross-Omicron immunogenicity of a Pfizer BA4/5 vs a Moderna BA1 bivalent COVID-19 vaccine in patients with renal disease on dialysis or post-transplant in Canada. The investigators found that both bivalent vaccines boosted cross-Omicron responses above baseline, but that there was no significant difference between the two different vaccines. Bivalent boosting in participants with evidence of prior SARS-CoV-2 infection led to the highest titers. Transplant patients did not respond well to either boosting regimen, particularly against recent, immune evasive XBB variants.

The strengths of this study are as follows:

- Takes advantage of the fact that in Canada, both a BA.1 and a BA.4/5 bivalent vaccine were authorized for use and can therefore be compared. (In the US, the BA.1 vaccines were not authorized).
- Prospective study
- Includes a novel population of patients with chronic kidney disease including a smaller cohort of renal transplant recipients
- Uses an extended panel of pseudoneutralization constructs with data for the most recent Omicron subvariants including XBB.
- Addresses immunogenicity in a relatively older cohort (median age 69 years)
- Controls for prior infection as measured by anti-NC seropositivity.

Weaknesses

- Observational
- Short follow up of 30 days
- Study population is not necessarily generalizable
- ***There is another randomized controlled trial looking at bivalent BA4/5 and bivalent BA1 immune responses with similar findings (Branche AR Clin Infect Dis. 2023;ciad209. doi:10.1093/cid/ciad209). This diminishes the impact of the study submitted here. The RCT was in a different population (not CKD patients) and not in a real-world setting, and didn't include the Pfizer vaccine. So there is a difference, but it needs to be noted***
- ***The study is comparing Pfizer BA4/5 to Moderna BA1, and there are differences in the mrna dose and platform immunogenicity that might be confounding results***

Thank you for summarizing the strengths and limitations of this study. We agree that our study has similarities to the randomized controlled trial noted by the reviewer which compares the immunogenicity of the Pfizer/BioNTech BA.1 versus BA.4/5 vaccines. The findings from the referenced study (Branche *et al.* Clinical Infectious Diseases 2023) are similar to those which we have reported in this study including that XBB.1.5 and BQ.1.1 neutralizing antibody titers were similar between BA.1 and BA.5 bivalent vaccines. However, there are a number of differences in the randomized trial which evaluated healthy adults aged 18-49 who had received a primary series with a single boost, and the BNT162b2 BA.1 bivalent vaccine was evaluated rather than

mRNA-1273 BA.1 vaccine. Furthermore, BQ.1.1 and XBB.1.5 neutralizing antibodies were only evaluated in a subset of participants in that study.

We have added this to our discussion as follows:

“Our study findings conducted in a real-world setting are similar to the findings of a randomized controlled trial in 202 individuals which compared the Pfizer/BioNTech BA.1 versus BA.4/5 vaccine and found similar neutralizing antibodies against BQ.1.1 and XBB.1.5 with either bivalent vaccine.¹ However, this randomized trial differed from our study as it only included healthy individuals and did not evaluate the mRNA-1273 BA.1 vaccine.”

- The study does not control for the fact that the Moderna BA1 population may be qualitatively different than the BA4/5 population. For example, Moderna BA1 was authorized in Canada a month before Pfizer BA4/5. The extended data shows that all the people who waited for Pfizer had gotten all their previous doses with Pfizer, whereas the Moderna group had more heterogenous prior vaccines. This can influence immunogenicity. Then there is the issue of early vs. late adopters (Moderna adopters were early, Pfizer bivalents were late); this may track with the frequency and nature of prior SARS-CoV-2 exposures and the subsequent character of immune responses following bivalent boost. These limitations should be discussed.

Thank you for this comment. We agree that due to the observational nature of this study there could be residual confounding between the two study groups although we made efforts to control several key confounders.

Because we have previously demonstrated that the original mRNA-1273 vaccine has greater immunogenicity in comparison to the original BNT162b2 vaccine (Yau *et al.* CMAJ 2022; 194 (8) E297-E305), we considered adjusting for the prior vaccine types in our linear mixed effects model, but due to the multiple vaccine types and variables already within the model we did not want to over-specify the model. We have now conducted an additional analysis adjusting for initial two dose and third vaccine dose types. Initial two dose vaccine types were collinear (as largely participants received the same first and second vaccine type) and therefore were combined in the model. By adjusting for initial vaccine types, the results were unchanged (with higher p-values as shown in the table below). This additional analysis adjusting for prior vaccine types supports no significant difference in neutralizing antibodies against all variants of concern by bivalent vaccine type:

Extended Data Table 6: Neutralizing antibody response differences between BNT162b2 BA.4/5 vaccine and mRNA-1273 BA.1 vaccine while accounting for initial two dose vaccine type, third vaccine dose type, anti-nucleocapsid status, number of vaccine doses, patient type, and anti-nucleocapsid positivity

Variant of Concern	p-value
Wild-Type	0.26
BA.1	0.49
BA.5	0.23
BQ.1.1	0.50
XBB.1.5	0.39

We agree that early-adopters may be qualitatively different from late adopters (e.g., more risk averse and may take additional precautions to avoid contracting COVID-19). Furthermore, due to temporal differences in vaccine availability of the mRNA-1273 BA.1 (approved in Canada September 1, 2022) and BNT162b2 BA.4/5 COVID-19 vaccines (approved in Canada October 7, 2022) the risk of COVID-19 could have differed. Nevertheless, in our study we are reassured that only one individual developed a new COVID-19 infection based upon anti-nucleocapsid seroconversion during follow-up, indicating that this was unlikely to have affected the underlying study conclusions.

We have elaborated upon our limitations in our discussion as follows:

“Due to the observational nature of this study, there may have been residual confounding. Individuals who chose the mRNA-1273 BA.1 vaccine, which was available earlier in Canada, were more likely to have received prior mRNA-1273 vaccination in comparison to those who received the BNT162b2 BA.4/5 vaccine. However, in a sensitivity analysis wherein we adjusted for initial three dose vaccine types, the underlying conclusions were unchanged. Due to temporal differences in mRNA-1273 BA.1 versus BNT162b2 BA.4/5 vaccine availability, it is possible that differences in COVID-19 infections could have occurred based upon epidemiologic and participant factors. However, only one individual contracted COVID-19 (as determined by anti-nucleocapsid seroconversion) during follow-up. As such, COVID-19 infections were unlikely to have affected the underlying study conclusions.”

- In Figure 2, there appear to be a lot of NC seroconversions between baseline and post-bivalent x 1 month. The methods say that baseline samples were collected ““prior to”“ bivalent dosing, and that the study started in July 2022 ““ so when were the baseline samples obtained, and was it the same for everyone? If there were collected right before bivalent dosing (say that morning), then the data would suggest that a lot of people had breakthrough infections in the 30 days post-vaccine. This definitely happens. But that could really diminish the ability to detect differences between the groups - because if both groups had lots of BA5 exposure then that might swamp the impact of the BA4/5 or BA1 vaccine.

We agree that anti-nucleocapsid seroconversion is an essential confounder to consider. In this study, “pre-bivalent” samples were collected immediately prior to vaccination, and follow-up samples were collected approximately 1 month later.

The overall incidence of new COVID-19 infections was very low during this study due to the short follow-up period (which we agree would otherwise diminish the ability to detect a difference between groups). As described above, we have presented extended data on anti-nucleocapsid at both baseline and follow-up and found only 1/98 participants had new anti-nucleocapsid seroconversion at 1 month follow-up making it unlikely that this was a key confounder of study findings. Furthermore, no clinical COVID-19 infections were reported during the study period (as diagnosed by RT-PCR or rapid antigen testing). The low rates of infection reflect the short follow-up period, high prevalence of prior infection, and epidemiology of COVID-19 infections in Ontario, Canada at the time.

- Transplant patients have a very different host immune status compared to hemodialysis patients, depending on the level of immunosuppression present. This is born out in the Figure 2c data which shows how poorly the transplant patients responded to vaccine. I'm not sure it makes sense to lump them together when looking at all the other analyses. Especially because I think the transplant patients were exclusively in the Moderna group as per the extended data.

We agree that transplant recipients are substantially more immunosuppressed than patients receiving dialysis due to their triple immunosuppressive medication regimen. For this reason, in **Figure 2** we demonstrated the significant differences between the hemodialysis and the transplant participants neutralizing antibody response. The inclusion of the transplant recipients (who all received the mRNA-1273 BA.1) vaccine would be expected to bias towards finding a difference in neutralizing antibody response between the mRNA-1273 BA.1 and BNT162b2 BA.4/5 vaccines which we did not observe; supporting that a clinically significant difference between the two available bivalent vaccines was not observed in this population.

We have also repeated our analyses in hemodialysis participants alone. The underlying conclusions were unchanged in this analysis as shown in **Extended Table 10**, and we have included this in the manuscript as a sensitivity analysis as follows:

“The lack of difference between vaccine types was unchanged when the analysis was restricted to only hemodialysis participants: wild-type (P=0.50), BA.1 (P=0.25), BA.5 (P=0.098), BQ.1.1 (P=0.14), or XBB.1.5 (P=0.13).”

We did not analyze the transplant recipients separately due to the lower numbers and because they exclusively received the mRNA-1273 BA.1 bivalent vaccine as the fifth vaccine dose, and therefore we could not evaluate differences by vaccine type in this population. The analysis restricted to hemodialysis patients is shown in the following **Extended Figure S3**.

Extended Data Fig 3 | Neutralizing antibodies against wild-type, BA.1, BA.5, BQ.1.1, and XBB.1.5 by bivalent vaccine type among hemodialysis participants (n=83). Solid red line indicates median level. Dots represent individual serum samples collected. There was no significant difference between neutralizing antibody levels by vaccine type for wild-type ($P=0.50$), BA.1 ($P=0.25$), BA.5 ($P=0.098$), BQ.1.1 ($P=0.14$), or XBB.1.5 ($P=0.13$).

This paper takes advantage of a unique opportunity to compare BA4/5 and BA1 bivalent vaccines in a real-world setting, with a special emphasis on patients with chronic kidney disease who are either receiving hemodialysis or post-transplant. The study adds to the literature showing that multi-valent vaccines may not need to exactly match a currently circulating strain in order to provide increased breadth. This has important implications for how governments and industry will invest in new COVID-19 vaccine updates. The paper's impact is diminished by an existing RCT that also compares BA1 vs. BA4/5 bivalent vaccines. Nevertheless, it shows that these results are borne out in a real world setting and in a population with high risk for progression to severe disease in the setting of breakthrough infection.

The paper would benefit from highlighting the special population being studied, such as in the title. There should be a greater discussion of the limitations inherent to an observational study, with a particular emphasis on more subtle ways that comparative groups can be different.

Both hemodialysis and kidney transplant recipients are defined as having end-stage kidney disease. As suggested by the reviewer, we have revised our title to highlight that this study focuses on hemodialysis and kidney transplant recipients as follows:

“Omicron variant BA.1, BA.5, BQ.1.1, and XBB.1.5 Neutralizing Antibodies Following BNT162b2 BA.4/5 versus mRNA-1273 BA.1 Bivalent Vaccination in Patients with End-Stage Kidney Disease”

In the discussion we have expanded upon the fact that a strength of this study is the focus on the dialysis and kidney transplant recipient population who would be considered a population with lower immunogenicity and at higher risk for severe COVID-19 outcomes as follows:

“This study also focuses on a vulnerable patient population as both kidney transplant recipients and individuals receiving hemodialysis have lower humoral responses to COVID-19 vaccination and are at higher risk for severe COVID-19 outcomes.^{2,3} Therefore, characterizing the dynamics of humoral response to bivalent vaccination will help to inform future vaccination strategies in vulnerable populations, including the timing of additional boosters. Similarly, a recent study in patients with hematologic malignancy demonstrated the utility of serologic testing in immunocompromised populations for predicting future risk of COVID-19 infection.⁴”

We have included a more fulsome discussion regarding the limitations of this observational study as follows:

“Due to the observational nature of this study, there may have been residual confounding. Individuals who chose the mRNA-1273 BA.1 vaccine, which was available earlier in Canada, were more likely to have received prior mRNA-1273 vaccination in comparison to those who received the BNT162b2 BA.4/5 vaccine. However, in a sensitivity analysis wherein we adjusted for initial three dose vaccine types, the underlying conclusions were unchanged. Due to temporal differences in mRNA-1273 BA.1 versus BNT162b2 BA.4/5 vaccine availability, it is possible that differences in COVID-19 infections could have occurred based upon epidemiologic and participant factors. However, only one individual contracted COVID-19 (as determined by anti-nucleocapsid seroconversion) during follow-up. As such, COVID-19 infections were unlikely to have affected the underlying study conclusions.”

Reviewers' Comments:

Reviewer #1:

Remarks to the Author:

The authors have perfectly replied to reviewers' comments. The manuscript has been improved.

Reviewer #2:

Remarks to the Author:

The revised manuscript has taken into consideration the reviewer comments and responded thoroughly and appropriately. I appreciate the addition of a number of new extended figures (for sensitivity analyses) as well as new text, addressing potential confounders. The updated title also reflects the paper's strength, e.g., of focusing on the vulnerable population of those with ESRD. My recommendation is to accept the paper for publication.